# SLITRK5 is a negative regulator of hedgehog signaling in osteoblasts

Jun Sun [1,8], Dong Yeon Shin[1,7,8], Mark Eiseman[1], Alisha R. Yallowitz[1], Na Li[2], Sarfaraz Lalani[1], Zan Li [1], Michelle Cung [1], Seoyeon Bok [1], Shawon Debnath[1], Sofia Jenia Marquez[1], Tommy E. White[3], Abdul G. Khan[3], Ivo C. Lorenz [3], Jae-Hyuck Shim [4], Francis S. Lee [5], Ren Xu[2 ✉] & Matthew B. Greenblatt [1,6 ✉]

Hedgehog signaling is essential for bone formation, including functioning as a means for the growth plate to drive skeletal mineralization. However, the mechanisms regulating hedgehog signaling specifically in bone-forming osteoblasts are largely unknown. Here, we identified SLIT and NTRK-like protein-5(*Slitrk5*), a transmembrane protein with few identified functions, as a negative regulator of hedgehog signaling in osteoblasts. Slitrk5 is selectively expressed in osteoblasts and loss of *Slitrk5* enhanced osteoblast differentiation in vitro and in vivo. Loss of SLITRK5 in vitro leads to increased hedgehog signaling and overexpression of SLITRK5 in osteoblasts inhibits the induction of targets downstream of hedgehog signaling. Mechanistically, SLITRK5 binds to hedgehog ligands via its extracellular domain and interacts with PTCH1 via its intracellular domain. SLITRK5 is present in the primary cilium, and loss of SLITRK5 enhances SMO ciliary enrichment upon SHH stimulation. Thus, SLITRK5 is a negative regulator of hedgehog signaling in osteoblasts that may be attractive as a therapeutic target to enhance bone formation.

[1] Department of Pathology and Laboratory Medicine, Weill Cornell Medicine, New York, NY, USA. [2] State Key Laboratory of Cellular Stress Biology, School of Medicine, Xiamen University, Xiamen, Fujian, China. [3] Tri-Institutional Therapeutics Discovery Institute, New York, NY, USA. [4] Division of Rheumatology, Department of Medicine, University of Massachusetts Medical School, Worcester, MA, USA. [5] Department of Psychiatry, Weill Cornell Medical College, New York, NY, USA. [6] Research Division, Hospital for Special Surgery, New York, NY, USA. [7] Present address: Research Center, LegoChem BioSciences, Inc., Daejeon, South Korea. [8] These authors contributed equally: Jun Sun, Dong Yeon Shin. ✉email: xuren526@xmu.edu.cn; mag3003@med.cornell.edu

Current treatments for osteoporosis all have major limitations, which includes rare but severe toxicities, limits on maximum duration of therapy, efficacy in certain anatomic sites, or inducing a low bone turnover state that is undesirable in some contexts, such as during repair of skeletal injury[1]. As many of these effects appear to be inherent to the molecular targets of these agents, ultimately addressing this issue will require identification of new therapeutic targets to increase bone formation. In this respect, it is notable that while the Hedgehog (Hh) pathway plays a fundamental role in bone development and homeostasis, it has yet to be therapeutically harnessed for skeletal disorders. The Hh pathway is initiated through the binding of Hh ligands to the transmembrane receptor Patched1 (PTCH1), relieving its repressive effects on Smoothened (SMO). Activated SMO moves to the primary cilium where it acts through several intermediates to ultimately activate GLI family member transcription factors[2]. GLI transcription factors, primarily GLI1 and GLI2, subsequently enter the nucleus to activate downstream target gene expression, which includes transcriptional feedback regulation of Hh pathway components themselves (*Ptch1*, *Gli1*, and *Hhip*)[3]. Disruption of Hh signaling leads to multiple bone defects. Ihh deficient mice display dwarfism with disruptions in growth plate structure[4]. *Ptch1* haploinsufficiency leads to increased bone mass in mice, which is driven by enhanced osteoblast responsiveness to *Runx2*[5]. Conversely, *Gli1* haploinsufficiency causes decreased bone mass and impaired fracture healing in mice[2,6]. In addition, mice with an upregulation of Hh signaling in mature osteoblasts display both increased bone formation and excess RANKL-driven osteoclastogenesis, which results in enhanced bone resorption and reduced bone mass[7]. Given the central role of the Hh pathway in osteoblasts, proper regulation of this pathway is essential. However, little is known about how the responses of osteoblasts to Hh ligands is "finetuned".

We performed an initial screen for transmembrane proteins displaying selective expression in osteoblasts, as this gene set will be enriched for novel receptors or co-receptors that may be druggable, and identified SLITRK5 as a co-receptor that regulates Hh signaling. The SLITRK family, composed of SLITRK1 through SLITRK6, are type I single pass transmembrane proteins. Their nomenclature is based on containing N-terminal extracellular leucine-rich repeat (LRR) domains, similar to those of Slit proteins, and an intracellular carboxyl terminus that has sequence similarity to neurotrophin receptors (Trks)[8]. While there have been relatively few studies of SLITRK family members, studies to date show that *Slitrks* are highly expressed in the central nervous system and play roles in neuronal survival, neurite growth and synapse formation[9–11]. In mice, genetic deletion of *Slitrk5* leads to defects in corticostriatal neurotransmission and obsessive–compulsive disorder-like behaviors[12]. Here we find that loss of Slitrk5 enhances osteoblast differentiation and function in vitro and in vivo by directly regulating Hh signaling in osteoblasts. Slitrk5 represses the expression of downstream Hh target genes by its direct interactions with Shh and Ptch1. Taken together, this identifies Slitrk5 as a novel Hh co-receptor that represses Hh signaling specifically in osteoblasts to regulate bone formation.

## Results

**Slitrk5-deficiency promotes osteoblastogenesis in vitro**. To identify druggable targets that may increase bone formation, we screened gene expression data for transmembrane proteins showing selective expression in osteoblasts[13,14]. *Slitrk5* was identified as specifically expressed in osteoblasts, apart from its robust expression in the nervous system, and it has only very modest expression in osteoclasts and bone marrow cells (Fig. 1a and Supplementary Fig. 1a). To confirm that SLITRK5 shows selective expression in osteoblasts, a reporter mouse with an insertion of a beta-galactosidase cassette into a *Slitrk5* intronic sequence was used[12]. Beta-galactosidase staining confirmed *Slitrk5* expression in osteoblasts residing on the trabecular bone surface and periosteum, showing colocalization of staining with the osteoblast marker osteopontin (OPN) (Fig. 1b and Supplementary Fig. 1b). To investigate the role of *Slitrk5* in osteoblast differentiation, calvarial osteoblasts isolated from WT and *Slitrk5$^{-/-}$* mice were cultured under osteoblast differentiation conditions. *Slitrk5$^{-/-}$* osteoblasts displayed enhanced differentiation and increased mineralization capacity as indicated by alizarin red staining (Fig. 1c). Likewise, an increase in alkaline phosphatase (ALP) activity was observed in *Slitrk5$^{-/-}$* osteoblasts (Fig. 1d). In addition, we measured the mRNA levels of osteoblast marker genes after 6 and 12 days of differentiation. Consistent with an increase in ALP and mineralization activity, expression of characteristic osteoblast transcripts, including *Runx2*, *Sp7*, *Bsp*, *Ocn*, and *Alpl*, were all markedly increased in *Slitrk5$^{-/-}$* osteoblasts (Fig. 1e). Thus, *Slitrk5* is selectively expressed in osteoblasts and *Slitrk5* represses osteoblast differentiation in vitro.

**Slitrk5 is a negative regulator of hedgehog signaling in osteoblasts**. A previous study showed that *Slitrk5* is critical for brain-derived neurotrophic factor (BDNF)-dependent signaling in neural cells[15]. To assess whether *Slitrk5* regulates osteoblast differentiation through BDNF signaling, WT and *Slitrk5$^{-/-}$* calvarial osteoblasts were treated with BDNF over the course of osteoblast differentiation. BDNF did not impact osteoblast differentiation in either WT or *Slitrk5$^{-/-}$* cells (Fig. 2a). In addition, WT or *Slitrk5$^{-/-}$* cells were treated with K252A, an inhibitor of the TRK family of receptor tyrosine kinases, that blocks signaling by BDNF and other TRK family ligands. TRK inhibition was associated with decreased, not increased, ALP levels (Supplementary Fig. 2a). Thus, *Slitrk5* likely controls osteoblast differentiation through an alternative mechanism.

A prior study reported that *Slitrk5* expression was significantly increased in a Hh-induced mouse model of medulloblastoma[16] (Supplementary Fig. 2b). Given that many Hh signaling components or regulators are themselves targets of Hh signaling, this raised the possibility that SLITRK5 regulates Hh activity. To explore if Hh activity is enhanced in a manner consistent with the augmented osteoblast differentiation in Slitrk5$^{-/-}$ cells, the expression of Hh target genes were examined in cultured osteoblasts. As shown in Fig. 2b, the level of these genes was upregulated in *Slitrk5$^{-/-}$* osteoblasts, suggesting that Hh signaling is augmented in the absence of SLITRK5. Knockdown of *SLITRK5* in Saos2 human osteoblast-like cells similarly increased the expression level of marker genes of Hh pathway activity, including *Gli1*, *Gli2* and *Ptch1* (Supplementary Fig. 2c). Conversely, over-expression of SLITRK5 had the opposite effect and suppressed responses to SHH, shown by decreased levels of the Hh reporter genes *Gli1*, *Ptch1*, and *Hhip* (Fig. 2c). Similarly, the activation of a *Gli1*-responsive reporter gene was suppressed by overexpression of *Slitrk5* (Fig. 2d). These data identify *Slitrk5* as a negative regulator of Hedgehog signaling in osteoblasts.

Activation of hedgehog signaling can induce osteoblast differentiation[17]. To investigate whether *Slitrk5*-deficient osteoblasts are more sensitive to SHH stimulation, WT and *Slitrk5$^{-/-}$* osteoblasts were treated with different doses of SHH and osteoblast differentiation was assessed by ALP activity. Loss of *Slitrk5* amplified the effect of SHH on osteoblast differentiation, effectively producing an approximately 8-fold increase in Hh ligand potency (Fig. 2e). Similarly, knockdown of *Slitrk5* in calvarial osteoblasts also increased SHH-induced ALP activity in

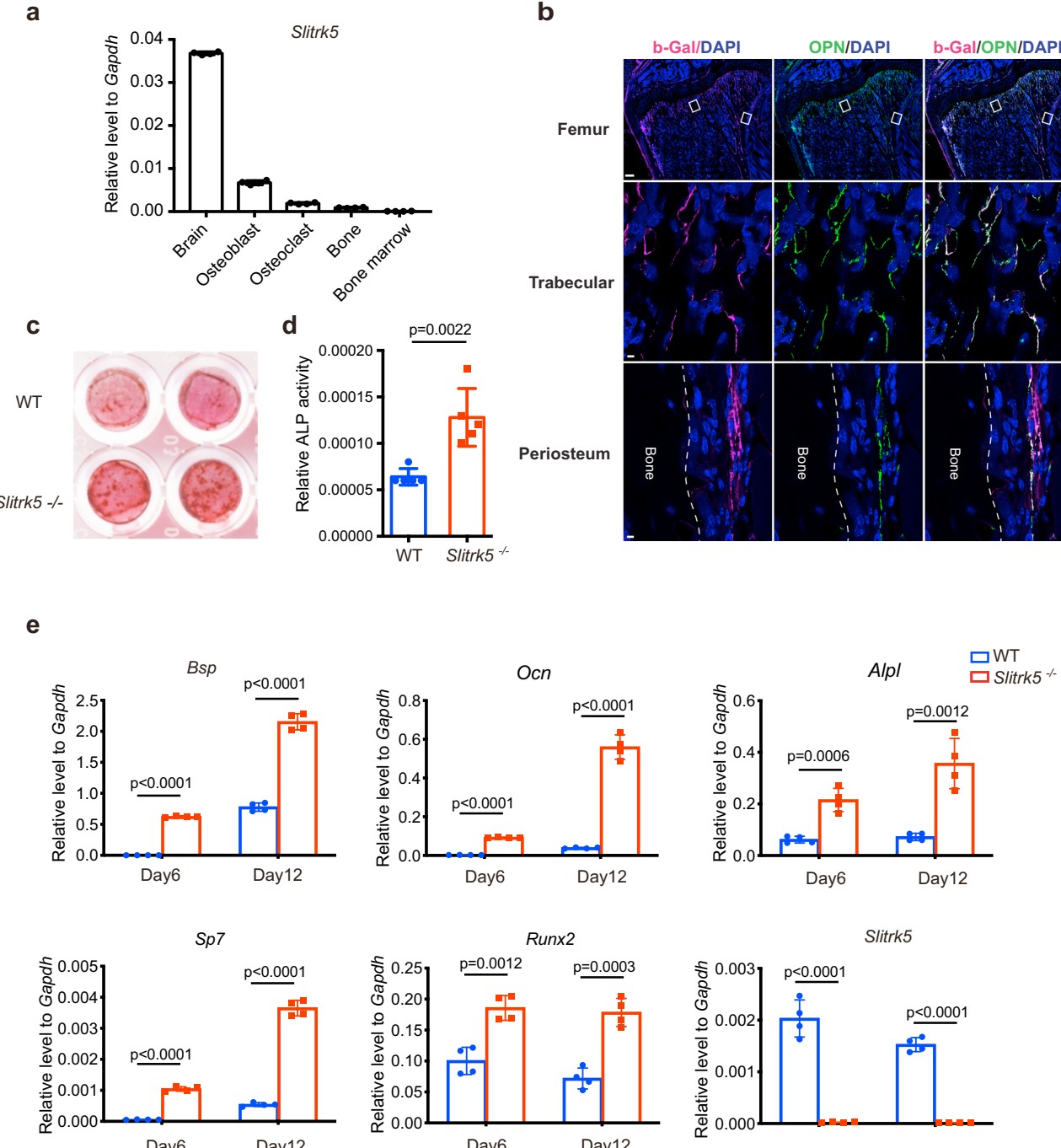

**Fig. 1 Slitrk5 is expressed in osteoblasts and negatively regulates osteoblastogenesis. a** Expression of *Slitrk5* mRNA in indicated cells or tissues. Data are presented as mean ± s.d. *n* = 4 biologically independent samples. **b** Immunofluorescence staining of mouse femur sections with anti-beta-galactosidase and OPN antibodies, demonstrating the expression of *Slitrk5* in osteoblasts. Data are representative of two independent experiments, scale bar = 200/10/ 10 μm. **c, d** Primary osteoblasts from WT and *Slitrk5*⁻/⁻ mice were cultured in osteoblast differentiation medium. Mineralization activity was assessed by alizarin red staining (**c**) at day 14 of differentiation. Alkaline phosphatase (ALP) activity was measured at day 8 of differentiation (**d**). Data in (**d**) are presented as mean ± s.d. *n* = 5 biologically independent samples, two-tailed unpaired *t* test. **e** RT-PCR analysis of *Slitrk5* and osteoblast marker genes expression at day 6 and day 12 of differentiation in WT and *Slitrk5*⁻/⁻ osteoblasts during differentiation. *n* = 4 biologically independent samples, two-tailed unpaired *t* test. Data are presented as mean ± s.d.

a manner that was responsive to the degree of knockdown (Fig. 2f, g). Thus, loss of *Slitrk5* leads to increased hedgehog responsiveness during osteoblast differentiation.

**SLITRK5 interacts with SHH and PTCH1.** Activation of the hedgehog signaling pathway involves Hh ligand binding to

PTCH1 and activation of SMO at the cell membrane which is followed by the activation of GLI inside the cell[3]. To delineate which components of the Hh signaling pathway are regulated by *Slitrk5*, we first treated WT and *Slitrk5*⁻/⁻ calvarial osteoblasts with purmorphamine, a direct SMO agonist, and osteoblast differentiation was assessed by ALP activity. While purmorphamine

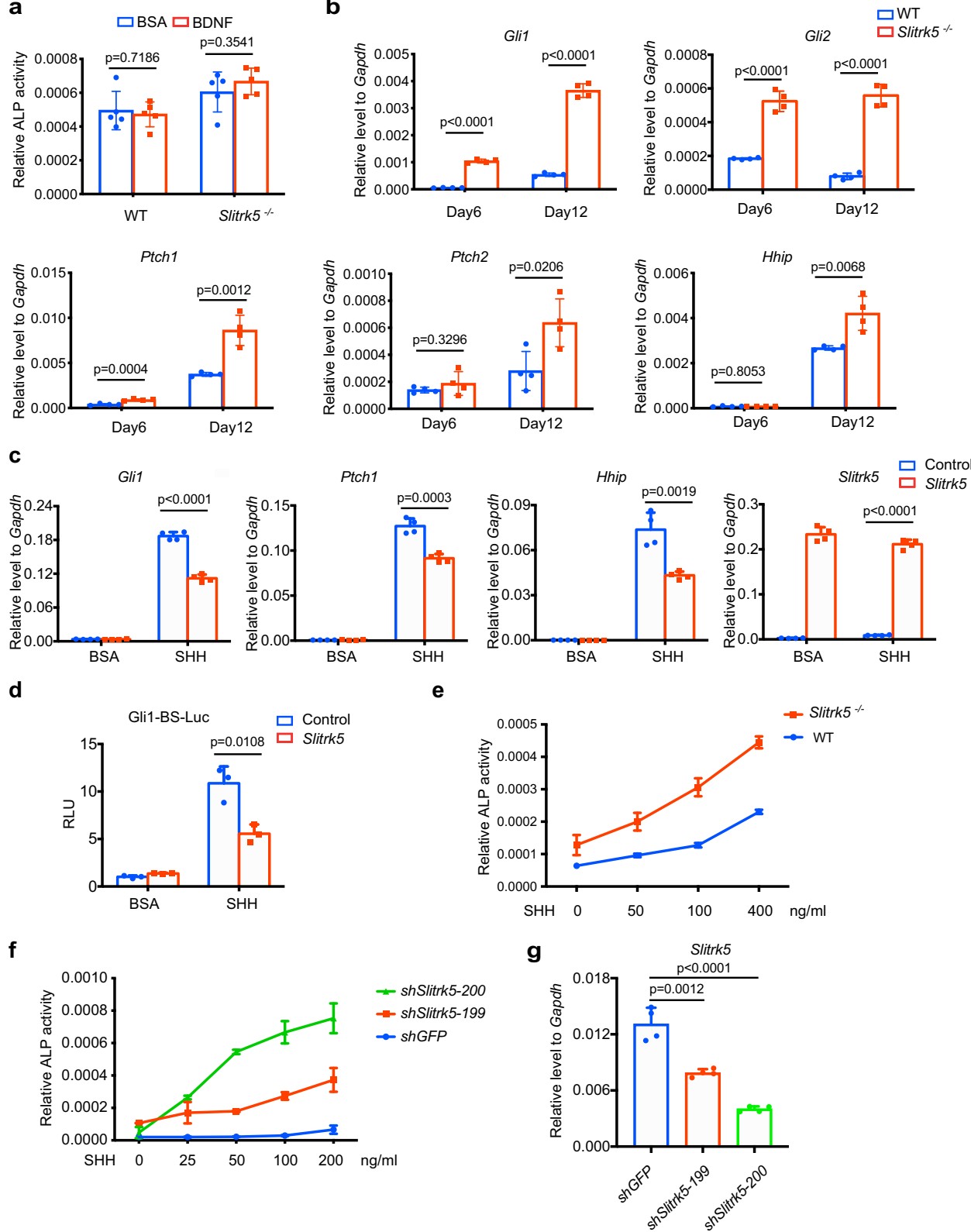

promoted osteoblast differentiation as expected, there was no difference in response between WT and *Slitrk5*$^{-/-}$ cells, indicating that SLITRK5 functions upstream of SMO, likely at the level of the PTCH1 complex (Fig. 3a and Supplementary Fig. 3a). As SLITRK5 is a transmembrane protein, we next investigated whether SLITRK5 binds to SHH, demonstrating an interaction between Flag-SLITRK5 and SHH in immunoprecipitation

assays (Fig. 3b). Among members of the SLITRK family, this ability to interact with SHH is restricted to SLITRK5, as SLITRK1 and SLITRK6 displayed no evidence of SHH interaction (Supplementary Fig. 3b, c). In keeping with this, SLITRK1 and SLITRK6 lacked the ability of SLITRK5 to suppress SHH responses in an enforced expression system (Supplementary Fig. 3d).

**Fig. 2 Slitrk5 is a negative regulator of hedgehog signaling in osteoblasts. a** Primary osteoblasts from WT and Slitrk5$^{-/-}$ mice were treated with BSA control or 40 ng/ml BDNF and cultured in osteoblast differentiation medium. ALP activity was measured at day 6 of osteoblast differentiation. Data are presented as mean ± s.d. $n = 5$ biologically independent samples, two-tailed unpaired $t$ test. **b** RT-PCR analysis of hedgehog signaling related gene expression at day 6 and day 12 of differentiation in WT and *Slitrk5*$^{-/-}$ osteoblasts. $n = 4$ biologically independent samples, two-tailed unpaired $t$ test. **c** C3H10T1/2 cells transfected with either control or Slitrk5 overexpression vectors were treated with BSA or 100 ng/ml SHH in serum-free medium for 48 h. Gli1, Ptch1, Hhip, and Slitrk5 mRNA levels were measured by RT-PCR. $n = 4$ biologically independent samples, two-tailed unpaired $t$ test. **d** C3H10T1/2 cells transfected with GLI1-luc/Renilla together with either control vector or *Slitrk5* overexpression vector were treated with BSA or 100 ng/ml SHH in serum-free medium for 36 h. Luciferase activity was measured to assess Hh signaling. $n = 3$ biologically independent samples, two-tailed unpaired $t$ test. **e** Primary osteoblasts from WT and *Slitrk5*$^{-/-}$ mice were treated with different doses of SHH and cultured in osteoblast differentiation medium. ALP activity was measured at day 8 of differentiation. Data are presented as mean ± s.d. $n = 5$ biologically independent samples. **f** Primary osteoblasts transduced with *Slitrk5* or *Gfp* shRNAs were treated with the indicated doses of SHH and cultured in osteoblast differentiation medium. ALP activity was measured at day 8 of osteoblast differentiation. Data are presented as mean ± s.d. $n = 2$ biologically independent samples. **g** Knockdown efficiency of *Slitrk5* was validated by qRT-PCR. $n = 4$ biologically independent samples, two-tailed unpaired $t$ test.

SLITRK5 is a single-pass transmembrane protein with an extracellular domain containing two LRR domains and intracellular carboxyl terminus. To further map the interaction region within SLITRK5, we generated a series of SLITRK5 truncation mutants, finding that a SLITRK5 construct lacking the entire extracellular domain, but not a construct lacking the first extracellular LRR domain alone, lost the ability to bind to SHH (Fig. 3c). Thus, SHH binds to either the second extracellular LRR domain or intervening linker sequences.

To address whether the interaction between SLITRK5 and SHH is direct, a cell-free binding assay was performed where a recombinant human SLITRK5 extracellular domain fragment was affixed to a solid phase and interaction with an epitope-tagged His-SHH was assayed via anti-His HRP (Fig. 3d). This approach revealed a direct interaction between SHH and the SLITRK5 extracellular domain. This was also confirmed by surface plasmon resonance which showed binding of a SLITRK5 extracellular domain fragment to SHH with a Kd of ~40 nM (Fig. 3e).

As these findings suggest that SLITRK5 may function as a SHH co-receptor, we examined if SLITRK5 may interact with the primary SHH receptor, PTCH1. Indeed, overexpressed HA-PTCH1 and Flag-SLITRK5 displayed a bidirectional, reciprocal interaction in immunoprecipitation assays in both HEK293 and C3H10t1/2 cells (Fig. 3f, g and Supplementary Fig. 3e). This interaction between SLITRK5 and PTCH1 was not affected by overexpression of SHH (Fig. 3f, g and Supplementary Fig. 3e). Using multiple SLITRK5 truncation mutants, the interaction between SLITRK5 and PTCH1 was mapped to the intracellular domain of SLITRK5 (Fig. 3h). Thus, SLITRK5 binds to SHH through its extracellular domain and to PTCH1 through its intercellular domain.

**SLITRK5 is located at the primary cilium and regulates SMO ciliary enrichment upon SHH stimulation.** Hedgehog signaling is functionally linked to the primary cilia in vertebrates. To investigate whether SLITRK5 is located at the cilium, we transduced primary osteoblasts with constructs encoding Flag-Slitrk5 and performed anti-Flag immunofluorescence. In line with a previous report[15], SLITRK5 showed a punctate distribution in the cytoplasm (Fig. 4a). Moreover, SLITRK5 additionally localized to the primary cilium, as indicated by the colocalization of the primary ciliary marker acetylated tubulin and SLITRK5 (Fig. 4a). SHH stimulation did not significantly increase the amount of SLITRK5 in the primary cilium (Fig. 4b). As ciliary localization of SMO and PTCH1 is linked to the activation of hedgehog signaling, we next examined whether SLITRK5 affects ciliary localization of SMO and PTCH1. Primary *Slitrk5*$^{-/-}$ and WT osteoblasts were infected with Smo-GFP or Ptch1-Flag virus and the ciliary traffic of SMO and PTCH1 was followed in response to SHH stimulation. The ciliary localization of SMO and PTCH1

was comparable in *Slitrk5*$^{-/-}$ and WT cells in the absence of SHH stimulation (Fig. 4c–f). Both *Slitrk5*$^{-/-}$ and WT osteoblasts also showed a similar decrease in the PTCH1 ciliary localization after SHH simulation. In contrast, SHH-induced SMO ciliary enrichment was enhanced in *Slitrk5*$^{-/-}$ cells (Fig. 4c–f). This enhancement in SHH-induced SMO ciliary recruitment is consistent with the overall enhanced SHH signaling seen the absence of SLITRK5 and with the results of purmorphamine stimulation indicating that SLITRK5 acts upstream of SMO, likely regulating signaling between PTCH1 and SMO.

**Slitrk5-deficiency promotes postnatal bone formation and fracture healing in mice.** Hedgehog signaling in osteoblasts plays an important role in the postnatal accrual of bone mass[18]. To examine whether regulation of hedgehog signaling by SLITRK5 observed in vivo is relevant to the regulation of bone mass in vivo, the expression of hedgehog signaling target genes was examined, finding increases consistent with enhanced hedgehog signaling activity in the tibias of *Slitrk5*$^{-/-}$ mice (Fig. 5a, b). To explore the role of *Slitrk5* in bone formation, dynamic histomorphometry analysis was performed on vertebrae from 7-week-old WT and *Slitrk5*$^{-/-}$ mice. The mineral apposition rate and bone formation rate in trabecular and cortical bone were both increased in *Slitrk5*$^{-/-}$ mice (Fig. 5c–f). Consistent with this, the number of osteoblasts on the bone surface was also increased in *Slitrk5*$^{-/-}$ mice (Fig. 5g, h). In addition, osteoclast numbers were increased in *Slitrk5*$^{-/-}$ mice (Supplementary Fig. 4a, b). In line with this concurrent increase in both bone formation and osteoclastogenesis, overall bone mass was not changed in *Slitrk5*$^{-/-}$ mice as assessed by micro-CT (Supplementary Fig. 4c, d), a finding also consistent with reports that enhanced hedgehog signaling in osteoblasts results in excessive bone resorption[7]. Thus, loss of *Slitrk5* in mice results in a high bone turnover state with increased bone formation and bone resorption.

Hedgehog also plays a critical role during fracture healing. SHH is expressed at fracture sites and regulates osteoblast proliferation and differentiation[19]. Activation of Hh signaling was reported to enhance fracture healing, raising the possibility that augmenting Hh signaling will improve skeletal repair[20]. To examine the role of *Slitrk5* in fracture healing, we performed femoral fractures in WT and *Slitrk5*$^{-/-}$ mice. The callus was harvested and analyzed by µCT 3 weeks post-fracture, finding that callus mineralization was increased in *Slitrk5*$^{-/-}$ mice (Fig. 5i, j). Accordingly, histology also showed increased bone in the callus of *Slitrk5*$^{-/-}$ mice (Fig. 5k). Consistent with the increased callus bone formation observed, *Col1a1* mRNA level was increased in the fracture callus of *Slitrk5*$^{-/-}$ mice, indicating enhanced osteoblast differentiation (Fig. 5l, m). Furthermore, the increased hedgehog signaling was observed in callus region of

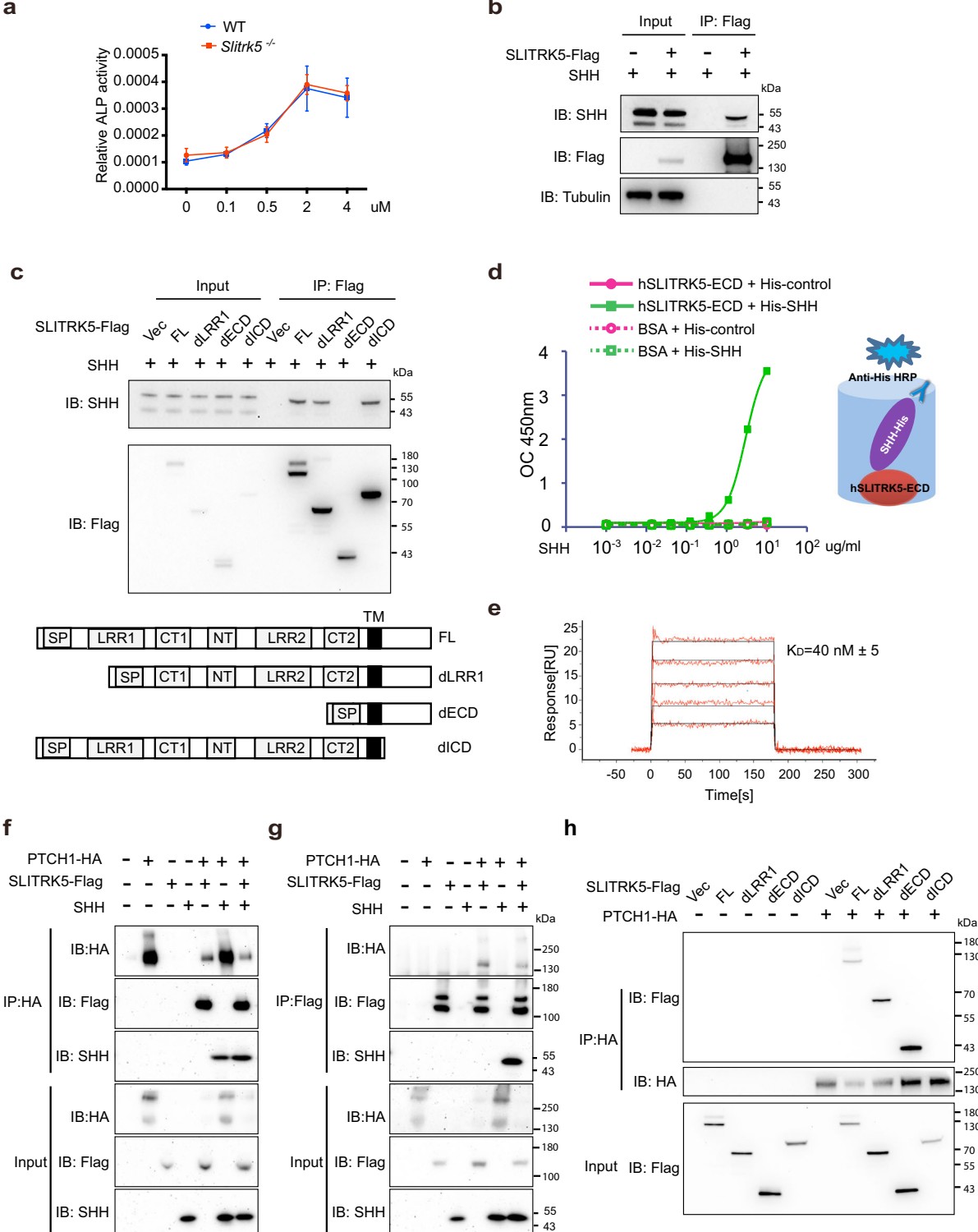

**Fig. 3 SLITRK5 interacts with SHH and PTCH1. a** Primary osteoblasts from WT and *Slitrk5*[−/−] mice were treated with the indicated doses of Purmorphamine and cultured in osteoblast differentiation medium. ALP activity was measured at day 3 of differentiation. Data are presented as mean ± s.d. $n = 6$ biologically independent samples. **b** Co-immunoprecipitation of Flag-SLITRK5 and SHH in HEK293T cells. Data are representative of four independent experiments. **c** Co-immunoprecipitation of SHH and Flag-tagged SLITRK5 truncation mutants in HEK293T cells. Data are representative of three independent experiments. **d** ELISA assay showing the interaction of hSLITRK5-ECD and SHH. **e** Surface Plasmon Resonance analysis of the binding of SHH to SLITRK5. SHH was injected over a SLITRK5 surface at 125, 62.5, 31.25, 15.625, and 7.8125 nM. **f, g** Co-immunoprecipitation of Flag-SLITRK5 and HA-PTCH1 in HEK293T cells with BSA or SHH treatment. Data are representative of two independent experiments. **h** Co-immunoprecipitation of PTCH1 and Flag-tagged SLITRK5 truncation mutants in HEK293T cells. Data are representative of three independent experiments.

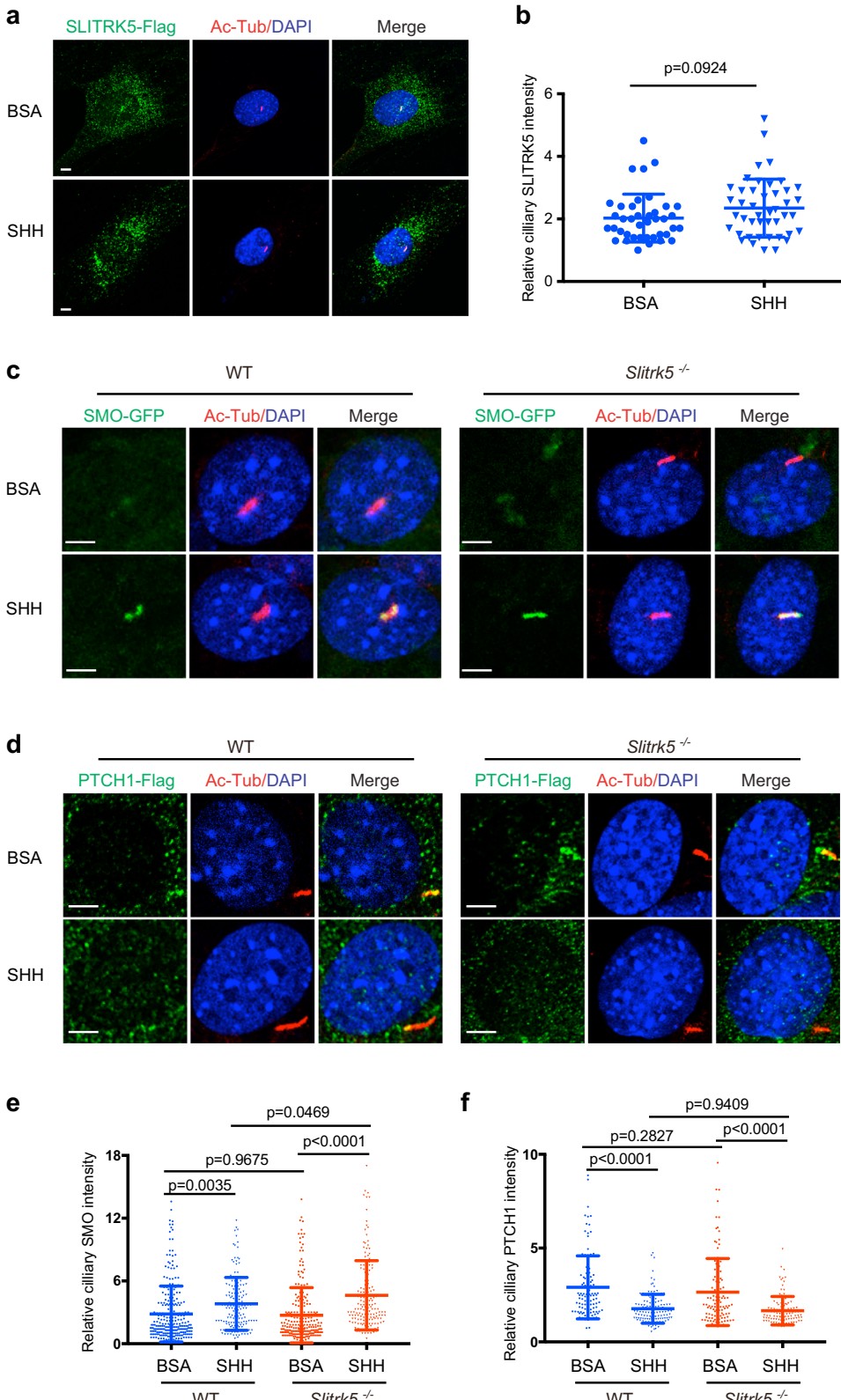

**Fig. 4 SLITRK5 regulates SMO ciliary enrichment upon SHH stimulation. a** Immunostaining for SLITRK5 (anti-Flag antibody, Green) and acetylated tubulin (Red) in osteoblasts treated with BSA or SHH for 4 h. Data are representative of three independent experiments, scale bar = 5 μm. **b** Quantification of the relative fluorescence intensity of ciliary SLITRK5-flag. n = 40–50 cells per group. Data are presented as mean ± s.d, two-tailed unpaired t test. **c**, **d** Ciliary localization of SMO-GFP (**c**) and PTCH1-Flag (**d**) in WT and Slitrk5⁻/⁻ osteoblasts treated with BSA or SHH for 4 h. Data are representative of three independent experiments, Scale bar = 5 μm. **e**, **f** Quantification of the relative fluorescence intensity of ciliary SMO-GFP (**e**) and PTCH1-Flag (**f**) in WT and *Slitrk5⁻/⁻* osteoblasts treated with BSA or SHH for 4 h. n = 100–300 cells per group. Data are presented as mean ± s.d, One-way ANOVA (P < 0.0001) followed by a Tukey's post hoc test.

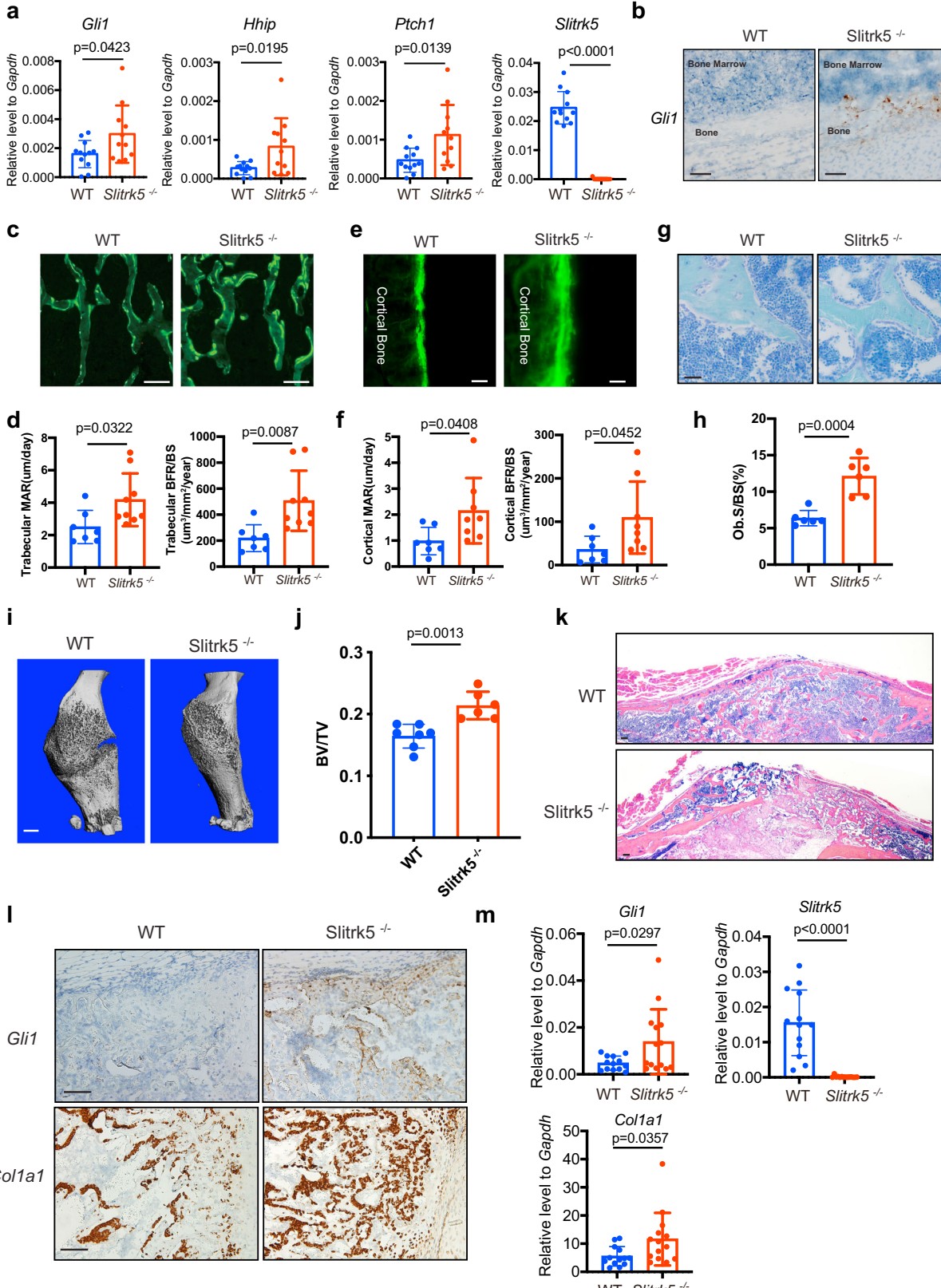

*Slitrk5*$^{-/-}$ mice as indicated by the upregulation of *Gli1* expression (Fig. 5l, m). Taken together, loss of Slitrk5 led to increased bone formation in both physiological conditions and enhanced fracture healing, phenotypes consistent with biochemical observations that SLITRK5 is a negative regulatory co-receptor in the Hh pathway in osteoblasts.

## Discussion

Hh signaling is progressively decreased as osteoblasts mature. This prevents excessive bone resorption as activation of Hh signaling in mature osteoblasts upregulates RANKL expression which drives osteoclastogenesis and bone resorption[7]. Osteoblast responsiveness to Hh ligands is thus tightly controlled by multiple

**Fig. 5 Slitrk5 deficient mice display increased bone formation. a** RT-PCR analysis of hedgehog signaling related gene expression in the tibias of 8-week-old WT and Slitrk5$^{-/-}$. $n = 11$ or 12 per group. Two-tailed unpaired $t$ test. **b** RNA in situ hybridization analysis using Gli1 probe in the tibias of WT and Slitrk5$^{-/-}$ mice at 4 day old. Data are representative of two independent experiments, scale bar $= 200\,\mu m$. **c**–**f** Calcein double labeling (**c**, **e**) and quantification of histomorphometric parameters (**d**, **f**) of the L3 vertebrae trabecular bone (**c**, **d**) and cortical bone (**e**, **f**) in 7-week-old WT and Slitrk5$^{-/-}$ female mice. Mineral apposition rate (MAR, µm/day), bone formation rate/bone surface (BFR/BS) (mm$^3$/mm$^2$/year). $N = 7$–9 per group. Data are presented as mean ± s.d, two-tailed unpaired $t$ test, data in (**c**, **e**) are representative of 7 (WT) or 9 (Slitrk5$^{-/-}$) independent samples. Scale bar $= 250\,\mu m$ (**c**) and $25\,\mu m$ (**e**). **g**, **h** Toluidine blue staining (**g**) and quantification of Ob.S/BS (**h**) of the L3 vertebrae in WT and Slitrk5$^{-/-}$ female mice at 7 week old. Osteoblast surface/bone surface (Ob.S/BS), $n = 6$ per group. Data are presented as mean ± s.d, two-tailed unpaired $t$ test, data in (**g**) are representative of six independent samples, scale bar $= 100\,\mu m$. **i** Representative µCT 3D images of mouse femurs at 21 days after open femoral midshaft fracture, scale bar $= 1\,mm$. **j** µCT measurement of BV/TV in callus area in WT and Slitrk5$^{-/-}$ mice at 21 days post-surgery. $N = 6$ or 7 per group. Data are presented as mean ± s.d, two-tailed unpaired $t$ test. **k** Representative H&E staining images of fracture callus from in WT and Slitrk5$^{-/-}$ mice at 21 days after an open femoral midshaft fracture. Data are representative of three independent experiments, scale bar $= 100\,\mu m$. **l** RNA in situ hybridization analysis using Gli1 and Col1a1 probes in the callus area of WT and Slitrk5$^{-/-}$ mice at 12 days post-surgery. Data are representative of two independent experiments, scale bar $= 100\,\mu m$. **m** RT-PCR analysis of Gli1, Col1a1, and Slitrk5 expression in the fracture callus from WT and Slitrk5$^{-/-}$ mice at 12 days after open femoral midshaft fracture, $n = 13$ or 14 per group. Two-tailed unpaired $t$ test.

levels of regulation. Gnas inhibits Hh signaling partially through activating the hedgehog inhibitor protein kinase A (PKA). Loss of Gnas increases Hh signaling, leading to enhanced osteoblast differentiation[21]. Another regulator named speckle-type POZ protein (Spop) is an E3-ubiquitin ligase adapter, regulating the ubiquitination and degradation of GLI3[22]. Spop-deficient osteoblasts showed increased GLI3 repressor levels and decreased Hh signaling, resulting in osteoblast differentiation defects[23]. Here, we identify Slitrk5 as a novel regulator of Hh signaling in osteoblasts. Slitrk5 acts at the most proximal steps in Hh signaling, acting upstream of SMO at the level of SHH and the PTCH1 receptor complex, enhancing SHH-induced osteoblast differentiation.

Here we identify SLITRK5 as a Hh co-receptor that acts at the level of the primary Hh receptor PTCH1 to regulate Hh signaling. SLITRK5, therefore, joins other Hh co-receptors including BOC, CDO, and GAS1[24–26]. Hedgehog interacting protein (HHIP) attenuates Hh signaling by competing with PTCH1 for binding to HH ligands[27–29]. HHIP is induced by Hh signaling, forming a negative regulatory feedback loop[27]. Similarly, Glypican-3 (GPC3) competes with PTCH1 for Hh ligand binding, inhibiting Hh signaling[30]. The binding affinity of SLITRK5 for SHH ($K_d \approx 40\,nM$) is similar to that of GPC3 for SHH ($K_d \approx 32\,nM$), arguing that SLITRK5 binds to Hh ligands at a physiologically relevant concentration. Both of these negative regulatory co-receptors have slightly lower affinity than that of positive regulatory co-receptors such as HHIP-SHH ($K_d \approx 6\,nM$)[29,30], which would suggest that with increasing ligand titration, first positive and then negative-regulatory pathways are engaged. This would allow productive signaling to be engaged first and then negative regulatory pathways are increasingly recruited with higher ligand concentrations. In addition, Slitrk5 is also transcriptionally upregulated by Hh signaling, suggesting that SLITRK5 forms a negative regulatory feedback loop in response to Hh stimulation. This also fits the overall pattern that many Hh pathway components, such as Gli1 and Ptch1, are themselves upregulated in response to Hh pathway activity.

Cdo and Boc displayed different expression patterns during mouse embryonic development and mice with mutation of Cdo exhibited holoprosencephaly while Boc mutants showed defective commissural axon guidance[26,31–33], suggesting the regulation of Hh signaling is cell context-dependent and Hh co-receptors may serve to provide tissue and context-specific "tuning" of Hh responsiveness. In line with this, we show here that, outside of the central nervous system (CNS), Slitrk5 expression is largely restricted to osteoblasts. Accordingly, Slitrk5$^{-/-}$ mice displayed increased bone formation but no other obvious Hh signaling related phenotypes outside of bone.

Given that Slitrk5 is also expressed in neural tissues, it is possible that Slitrk5 also modulates Hh signaling in neural cells and is therefore relevant to the oncogenesis of Hh pathway driven CNS tumors such as medulloblastoma. Supporting this, Slitrk5 expression is upregulated in tumors marked by active Hh signaling, such as Hh dependent medulloblastoma[16]. Similarly, missense mutations in SHH and Slitrk5 have been implicated in the attention deficit hyperactivity disorder[34,35], raising the possibility that SLITRK5-regulation of Hh signaling contributes to disease processes outside of bone.

There are currently very limited therapeutic drug options for bone repair, thus it is notable both that prior studies identify Hh as a key pathway regulating fracture repair[36,37] and that this study nominates SLITRK5 as a candidate therapeutic target relevant to this context. Hh signaling is activated during fracture healing and expression of a constitutively active SMO mutant in Col1-cre targeted osteoblasts led to enhanced bone formation in the fracture callus, suggesting that augmented Hh signaling can promote fracture healing[19,36]. A decrease in total Hh signaling activity has also been implicated as contributing to the age-related declines in fracture healing capacity and local osteoblast generation[37]. In addition, the hedgehog agonist purmorphamine enhanced bone regeneration in a calvarial defect mouse model[20,36]. Meanwhile, we here report that deletion of the negative Hh regulator Slitrk5 improved bone fracture healing. Thus, Hh signaling may have therapeutic importance for fracture healing. Given the narrow and osteoblast selective expression pattern of Slitrk5 alongside the absence of Hh associated phenotypes in nonskeletal tissues, targeting SLITRK5 is promising for offering an osteoblast-specific means to modulate Hh responses to enhance osteoblast differentiation and skeletal repair.

## Methods

**Mice.** Slitrk5$^{-/-}$ mice were previously described[12]. All experiments were performed according to the guidelines approved by the Animal Care and Use Committee of the Weill Cornell Medical College. All relevant ethical regulations have been complied with for animal testing and research. All mice were maintained in 12/12 light/dark cycle at room temperature of 20.5–22.5 °C and humidity of 30–70% and had ad libitum access to dry laboratory food and water.

**ELISA assay.** hSLTRIK5-ECD (2 µg/ml) or BSA (2 µg/ml) was added to the 96-well microtiter plates. The plates were incubated at room temperature for 2 h before being washed four times. SHH-His (1314-SH-025, R&D systems) or control His peptide was added and the plates were incubated for another 2 h at room temperature. After 4 times wash, the plates were incubated at room temperature for 30 min. After 4 times wash, substrate solution was added, and Relative Light Units was measured at 450 nM using a luminometer.

**Surface plasmon resonance.** Surface plasmon resonance experiments were performed using a Sierra Sensors MASS-1 (Bruker Daltonics, Billerica, MA).

Human SLITRK5 (2587-SK, R&D Systems, Minneapolis, MN) was immobilized on a research-grade high-capacity amine sensor chip (Bruker Daltonics, Billerica, MA) using the amine coupling method. SHH was diluted in PBS-T (135 mM sodium chloride, 2.7 mM potassium chloride, 4.3 mM Sodium Phosphate, 1.4 mM potassium phosphate, 0.05% Tween-20, pH 7.4) and flowed over the surface at a rate of 10 μl/min.

**Real time PCR analysis**. For analysis of gene expression, total RNA from cultured cells or tissues was extracted using TRIzol(Qiagen) according to the manufacturer's instructions. cDNA was then obtained using a High-Capacity cDNA Reverse Transcription Kit (Invitrogen). An SYBR Green polymerase chain reaction (PCR) Master Mix Kit (Applied Biosystems) was used for real-time PCR. QuantStudio 6 Flex RT-PCR Software v1.3 was used for mRNA analysis. Sequences of the PCR primers used are shown in Supplementary Table 1.

**Luciferase reporter assay**. C3H10T1/2 cells were transiently transfected with GLI1 responsive reporter plasmid (GLI1-BS-Luc) and Renilla luciferase plasmid together with control or Slitrk5 plasmid. Twenty-four hour after transfection, cells were treated with vehicle or SHH in serum-free medium for 36 h. Cells were then lysed and luciferase activity was measured using the Dual-luciferase reporter assay system (Promega).

**Histomorphometry**. Mice were injected with 20 mg/kg calcein (Sigma) at 10 days and 2 days before scarification. Plastic embedding, TRAP staining, and toluidine blue staining of the undecalcified lumbar region were performed as previously described[38]. Static and dynamic histomorphometric analyses were performed using the Osteomeasure Analysis System (Osteometrics).

**RNA in situ hybridization**. To detect Gli1 and Col1a1 RNA in formalin-fixed, paraffin-embedded tissues, ISH was performed using the BOND RNAscope Detection Reagents kit (DS9790, Leica Biosystems, Buffalo Grove, IL, USA) according to the manufacturer's instructions. Briefly, 20 ZZ probe pairs targeting the Gli1 and Col1a1 mRNA were designed and synthesized by Advanced Cell Diagnostics (catalog numbers 311008 and 537048). Tissue samples were incubated with Leica Epitope Retrieval 2 for 20 min at 95 °C, then pre-treated using Leica Protease at 40 °C for 20 min, then incubated with Gli1 or Col1a1 RNA probe at 40 °C for 240 mins. ACD AMP 1–6 was applied at 40 °C for 60–120 min for signal amplification before application of 3,3′-Diaminobenzidine (DAB).

**Histology and immunostaining**. Hind limbs from mice were dissected and fixed in ice-cold 4% paraformaldehyde solution for overnight and then decalcified in 0.5 M EDTA solution at 4 °C for 2 weeks. Samples were either embedded in optimal cutting temperature compound (Leica) and sectioned at 20 μm thickness or embedded in paraffin and cut into 5-μm-thick sections. Immunostaining for β-galactosidase and OPN was performed using an anti-β-galactosidase antibody (GTX77365; GeneTex) and an anti-OPN antibody (AF808, R&D Systems). For cilium staining, bone marrow stromal cells were harvested from 3-week-old mice and osteoblast differentiation was induced with ascorbic acid and β-glycerophosphate. Cells were then infected with lentiviral constructs encoding Slitrk5-flag, Ptch1-flag, or Smo-GFP. Cells were cultured under serum starvation conditions (medium with 0.5% fetal bovine serum (FBS)) for 48 h before 4 h SHH (300 ng/ml) treatment. After SHH treatment, cells were fixed with 4% paraformaldehyde (PFA) for 10 min, permeabilized with 0.1% (v/v) Triton X-100 for 10 min, blocked with 5% for 1 hr. Antibodies were used as follow: rabbit anti-Flag (1:500, cell signaling technology, 2368S) and mouse anti-acetylated tubulin (1:1000, Sigma, T7451). Imaging was performed with a Zeiss LSM 880 with an Airyscan high-resolution-detector confocal microscope. Carl Zeiss Zen 2.3 SP1 FP3 (black, v14.0.18.201, Germany) was used for immunofluorescence imaging analysis. To quantify the relative fluorescence of protein in the primary cilia, the CiliaQ ImageJ plugins(CiliaQ-0.1.4,CiliaQ Editor_JNH-0.1.0 and CiliaQ Preparator_JNH-0.1.0) were used for image segmentation and ciliary fluorescence quantification[39]. For each cell, the ciliary fluorescence intensity was normalized to cell body intensity.

**Osteoblasts culture and differentiation assays**. Primary osteoblasts were isolated from 5–7 days old mice by triple collagenase/Dispase II digestion. Cells were cultured in α-MEM medium (Gibco) containing 10% FBS, 2 mM L-glutamine, 1% penicillin/streptomycin, 1% HEPES, and 1% nonessential amino acids, and differentiated with ascorbic acid and β-glycerophosphate. For ALP activity, osteoblast number was assessed quantitatively using Alamar blue assay. Cells were then washed, fixed in 10% neutral-buffered formalin, and incubated with a solution containing 6.5 mM $Na_2CO_3$, 18.5 mM $NaHCO_3$, 2 mM $MgCl_2$, and phosphatase substrate (Sigma-Aldrich). ALP activity was measured by a luminometer. For extracellular matrix mineralization staining, cells were fixed in 10% neutral-buffered formalin and stained with alizarin red.

**Immunoprecipitation and immunoblotting**. HEK293T cells or C3H10T1/2 cells were transfected with the indicated DNA plasmids. Cells were lysed in RIPA buffer or TNT lysis buffer [10 mM Tris, 50 mM NaCl, 5 mM EDTA, 2 mM NaF, 30 mM sodium pyrophosphate, 100 mM $Na_3VO_4$, 0.5 mM PMSF, 1 μg/ml leupeptin, 5 μg/ml aprotinin, and 1% Triton X100]. Flag or HA agaroses were used in the immunoprecipitation and immunoprecipitated proteins were then subjected to sodium dodecyl sulfate-polyacrylamide gel electrophoresis and transferred to Immobilon-P membranes (Millipore, Billerica, MA, USA). The membranes were then blocked with 5% nonfat milk and incubated with specific antibodies.

**Antibodies and reagents**. BDNF (248-BD-005/CF, R&D systems) was used at a concentration of 40 ng/ml. Purmorphamine (540220-5MG) was purchased from EMD Millipore K252a (11338) was purchased from Cayman. SHH (8908SH) was purchased from R&D systems. EZview™ Red Anti-HA Affinity Gel (E6779, Sigma), EZview™ Red ANTI-FLAG® M2 Affinity Gel (F2426, Sigma), Flag peptide (F3290, Sigma) and HA peptide (I2149, Sigma) were used for immunoprecipitation experiments. The following primary antibodies were used: anti-acetylated tubulin (Sigma, T7451, clone 6-11b-1,0000106162, 1:1000), anti-tubulin (Santa Cruz, sc-23948,H160,C2316, 1:5000), anti-HA (cell signaling technology, 3724, C29F4, 12/2017, 1:2000), anti-DYKDDDDK (cell signaling technology, 14793s, D6W5B, 11/2018, 1:2000), anti-GFP (Abcam, ab13970, GR89472-25,1:5000), anti-β-galactosidase antibody (GTX77365; GeneTex,1:100), anti-SHH (H-160, sc-9024, Santa Cruz,1:1000), anti-SHH (E1, sc-365112, Santa Cruz,1:1000), anti-Osteopontin (AF808-SP, R&D systems1:400) and anti-Flag (9696S, Cell Signaling Technology, 1:1000). Cell lines were purchased from ATCC: C3H10T1/2 cells clone 8 (CCL-226), 293T (ATCC® CRL-3216™), Saos-2 (ATCC® HTB-85™).

**Bone fracture model**. Surgery was performed under isoflurane (2.5%) anesthesia via nosecone. The surgical sites were sterilized, and an incision was made on the right femur. The fracture was made using a Dremel saw with a diamond thin cutting wheel (Cat. #100230-724; VWR) and a 25-gauge syringe needle was inserted from femoral condyles to stabilize the femur. Muscles were reapproximated, and the skin was closed using wound clips.

**MicroCT analysis**. The mouse femurs were skinned and fixed in 70% ethanol. Femurs were scanned using a Scanco Medical uCT 35 system with a spatial resolution of 7 μm. The X-ray tube potential of 55 kVp, an xray intensity of 0.145 mA, and an integration time of 600 ms. For analysis of femoral bone mass, a region of trabecular bone 2.1 mm wide was contoured, starting 280 microns from the proximal end of the distal femoral growth plate. For analysis of fracture callus bone volume, the whole callus was contoured.

**Statistics**. All results are presented as the mean ± SD. Comparisons between two groups were analyzed using a two-tailed, unpaired Student's $t$ test. One-way Anova followed by Tukey's post hoc test was used when the data involves multiple group comparisons. GraphPad PRISM v.8.4.3 was used for statistical analysis.

**Reporting summary**. Further information on research design is available in the Nature Research Reporting Summary linked to this article.

## Data availability
All relevant data are available from the authors. Uncropped blots are provided in Supplementary Fig. 5. Source data are provided with this paper.

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

## Acknowledgements

This project was supported by a Career Award for Medical Scientists from the Burroughs Wellcome Fund and the NIH under awards DP5OD021351 and R01AR075585. This publication is based on research supported by the Pershing Square Sohn Cancer Research Alliance via an award to M.B.G. R.X. is supported by the National Key R&D Program of China (2020YFA0112900) and the National Natural Science Foundation of China (81972034 and 92068104 to R.X.). We thank Pathology Core Facilities at Weill Cornell Medicine for their technical assistance.

## Author contributions

Conception and design: M.B.G., R.X., Y.S. and J.S.; development of methodology and acquisition of data: R.X., J.S., D.Y.S., M.E., A.R.Y., N.L., S.L., S.B., S.D., T.E.W., A.G.K. and I.C.L.; analysis and interpretation of data: M.B.G., R.X., D.Y.S. and J.S.; writing and revision of the paper: M.B.G., R.X., J.S., A.R.Y., Z.L., M.C., S.J.M., F.S.L. and J.H.S.; study supervision: M.B.G.; All authors reviewed and gave final approval to the paper.

## Competing interests

J.H.S. is a scientific co-founder of AAVAA therapeutics and holds equity in this company. The remaining authors declare no competing interests.
