## [Peer Review File · Nature Communications]

Reviewers' Comments:

Reviewer #1:

Remarks to the Author:

In the manuscript entitled, "SLITRK5 is a novel regulator of hedgehog signaling in osteoblasts", Sun and Shin et al., provide evidence that the transmembrane protein SLITRK5 antagonizes Hedgehog signaling in osteoblasts. Specifically, the authors identify SLITRK5 as a novel component of the Hedgehog signaling pathway that interacts with both SHH and PTCH1, but does so via different domains (extracellular for SHH, intracellular for PTCH1). Overall, this work will be of high interest to the Hedgehog community, as well as to scientists studying bone development, and researchers investigating the regulation of cell signaling. However, as outlined below, there are a number of significant experiments that the authors should conduct to further validate their findings, as well as some important controls that are missing from their current study, which need to be included to confirm their results.

Major Comments:

1. In Figure 1B, the images are of too low resolution to determine if osteoblasts express Slitrk5 (this appears to be an issue with the PDF, since the labels are also low resolution). Can the authors co-stain with a Beta Galactosidase antibody and an osteoblast marker to confirm Slitrk5 expression?
2. I am very confused by the apparently contradictory data in Figure 1D (which shows significantly increased ALP activity in Slitrk5 mutant cells), and Figure 2A (which shows no significant changes in ALP activity in WT vs. Slitrk5 mutant cells). In reading the figure legends, there may be some difference in the length of the assay between these experiments, but I could not confirm this. Here, the authors need to clarify their experimental conditions between experiments, and also explain the discordance in their results.
3. Related to Figure 2A, if Slitrk5 is acting through the BDNF pathway, wouldn't you want to inhibit BDNF signaling and ask if this increases ALP activity? Perhaps a Trk inhibitor would be a better test of this hypothesis. This may be outside the scope of the current study, but this experiment makes more intuitive sense to me than adding exogenous BDNF.
4. The increase in Gli1 and Gli2 expression in Slitrk5^{-/-} cells (Figure 2B) is particularly interesting. Here, it would be important to examine other HH targets (e.g., Ptch1 and Ptch2), as the authors do in Figure 2C.
5. In Figure 3A, I have the same concern as in Figure 2A- why is there no difference in ALP activity between WT and Slitrk5 mutant cells? I was expecting that the results portrayed in Figure 1 would hold true across experiments.
6. The IP experiments in Figure 3B do not make sense to me. That is, why is there a band in the anti-FLAG input blot (lane one, top blot) in lysates that have not been transfected with SLITRK5-FLAG? Other issues in this blot (and the other blots) include the lack of MW markers, and the lack of a loading control for the input lanes.
7. The data with Slitrk6 presented in Figure S1B, C are very interesting. However, I would have liked the authors to explore this in greater detail. For example, does Slitrk6 expression inhibit expression of HH targets? What if you expressed Slitrk1 (or another Slitrk member)? In other words, do SLITRK proteins perform overlapping functions, or is this role in HH signaling restricted to Slitrk5? The authors do not need to exhaustively explore this possibility, but it seems that with a couple straightforward overexpression experiments (and/or IP experiments), the authors could greatly improve the impact of their work (in this case, either answer is an interesting one!).
8. The data presented in Figure 4 are quite interesting. However, there is no connection to changes in the level of HH signaling. If the authors could perform an assessment of HH pathway activity (e.g., Gli1 in situ hybridization or RNAscope, qPCR or western blot), that would significantly improve the connection with the previous figures.

Minor Comments:

1. In Figure 1A, D, E, as well as Figure 2A-D, the authors need to include individual data points in their histograms (as they do in Figure 4).
2. In Figures 1, 2, and S1, the authors vacillate between GAPDH, HPRT and beta-actin for normalization in their qPCR experiments. It would be helpful to include a sentence of rationale for changing these controls between experiments.
3. In Figure S3, the authors need to re-run their input so that the reader can assess the levels of dECD expression. This was not an issue in Figure 3C, so I am not sure why it is an issue here.
4. In Figures 3F vs. 3G, why is there a doublet in one FLAG blot (3G), but not the other (3F)?
5. In Figure S3, can the authors demonstrate that the deltaICD construct still interacts with SHH? If the authors can perform this experiment, and clean up their loading control, then I would recommend moving the revised figure to the primary data.

Reviewer #2:

Remarks to the Author:

In this manuscript by Sun et al., the authors reported that a membrane protein SLITRK5 is a previously unknown regulator of osteoblast differentiation and Hedgehog signaling inhibitor. The authors claim that SLITRK5 is expressed specifically in osteoblast cells and found that bone formation was increased in the *Slitrk5*^{-/-} mice. They further found that loss of *Slitrk5* resulted in enhanced Hedgehog (Hh) signaling, while *Slitrk5* overexpression inhibited Hh signaling. They showed that *Slitrk5* bound Shh ligand through the extracellular domain and *Ptch1* through the intracellular domain. The findings are interesting. However, several key points of *Slitrk5* expression and function in Hh signaling were not adequately addressed.

Major points:

1. The claim that *Slitrk5* is specifically expressed in osteoblast cells is very important for the conclusion and approached they used in this study. However, there is no convincing data to show the temporospatial expression patterns of *Slitrk5* in skeletal formation. In situ hybridization is the best way to achieve this goal. Figure 1a is an RT-PCR data with no cellular resolution. For instance, it is not known whether *Slitrk5* is expressed in periosteum or perichondrium or growth plates.
2. The data quality of Fig. 1b is poor. It looks like that beta-galactosidase expression was everywhere in the bone marrow of both wt and *Slitrk5*^{+/-} mice. Images of the entire bone with growth plates are necessary to support the claim that *Slitrk5* is expressed specifically in the osteoblast cells. The age of the mice where the cells or femur samples are taken must be indicated too.
3. Fig. 2, alteration of Hh signaling should be shown in vivo in the bone samples.
4. Fig. 3F, G, it looks like *Slitrk5* reduced the protein levels of PTCH1-HA levels. Is this related to *Slitrk5* function in regulating Hh signaling?
5. It looks like that *Slitrk5* did not interfere with the SHH-PTCH1 binding, as SHH regulates PTCH1 and SMOOTHED cilium localization, the cilium localization of *Slitrk5* and the impact of *Slitrk5* loss on cilium localization of PTCH1 and SMOOTHED should be investigated to gain mechanistic understanding of the inhibitory mechanism of *Slitrk5* in Hh signaling.
6. Fig. 4A, both cortical and trabecular bone formation must be shown.
7. Fig. 4G, osteoblast differentiation and Hh signaling should be investigated in vivo with immunohistochemistry and qRT-PCR
8. Fig. S4C, both cortical and trabecular bone perimeters must be shown.

Reviewer #1 (Remarks to the Author):

Major Comments:

1. In Figure 1B, the images are of too low resolution to determine if osteoblasts express Slitrk5 (this appears to be an issue with the PDF, since the labels are also low resolution). Can the authors co-stain with a Beta Galactosidase antibody and an osteoblast marker to confirm Slitrk5 expression?

We agree that improving the data supporting expression of SLITRK5 in osteoblasts would strengthen our conclusions. In the revised manuscript, we have replaced Figure 1b in the original submission with higher resolution versions and have attempted to avoid resolution drops occurring during pdf processing prior to submission. As suggested, we have also co-stained with a β -galactosidase and an osteoblast marker (osteopontin, OPN) by immunofluorescence and found clear co-localization of the two. We have added the new data as Figure 1b and have moved the prior immunohistochemical anti- β -galactosidase staining to Supplementary Figure 1b.

2. I am very confused by the apparently contradictory data in Figure 1D (which shows significantly increased ALP activity in Slitrk5 mutant cells), and Figure 2A (which shows no significant changes in ALP activity in WT vs. Slitrk5 mutant cells). In reading the figure legends, there may be some difference in the length of the assay between these experiments, but I could not confirm this. Here, the authors need to clarify their experimental conditions between experiments, and also explain the discordance in their results.

We agree that the clarity of our presentation of the ALP phenotype should be improved to avoid confusion. Yes, as the reviewer suggests, Figure 1d was performed at 8 days of differentiation whereas the ALP expression studies in Figure 2 were performed at 6 days of differentiation. In general, the later the differentiation time point examined, the more robust of a difference is observed in ALP levels between WT and Slitrk5^{-/-} osteoblasts. For studies examining the impact of SLITRK5 loss on hedgehog or BDNF signaling (such as Figure 2a or Figure 3a), we deliberately chose earlier time points to minimize any potential confounding that could arise due to basal differences in ALP between WT and Slitrk5-deficient groups complicating interpretation of differences in SHH stimulation-induced ALP activity.

3. Related to Figure 2A, if Slitrk5 is acting through the BDNF pathway, wouldn't you want to inhibit BDNF signaling and ask if this increases ALP activity? Perhaps a Trk inhibitor would be a better test of this hypothesis. This may be outside the scope of the current study, but this experiment makes more intuitive sense to me than adding exogenous BDNF.

We agree that additional data would strengthen conclusions that SLITRK5 is not exerting its effects in osteoblasts via alterations in BDNF pathway signaling. As suggested, we have treated

WT and *Slitrk5*^{-/-} osteoblasts with the TRK inhibitor K252A, finding that inhibition of BDNF signaling is associated with decreased, not increased, ALP levels (Supplementary Figure 2a). These findings are overall consistent with published studies on TRK pathway signaling in osteoblasts (Mikami et al. Differentiation 2012, Akiyama et al. Differentiation 2014; Johnstone et al. J Musculoskelet Neuronal Interact 2019). Overall, the direction of effect of BDNF and TRK family signaling on osteoblasts is either, depending on the experimental conditions and system used, minimal or opposite what would be expected if the effects of SLITRK5 on osteoblast differentiation were related to the function of SLITRK5 in regulating BDNF signaling described in Song et al. Dev Cell 2015. Thus, these experiments further justify the search for alternative mechanisms of SLITRK5 action in osteoblasts undertaken in the rest of this manuscript.

4. The increase in *Gli1* and *Gli2* expression in *Slitrk5*^{-/-} cells (Figure 2B) is particularly interesting. Here, it would be important to examine other HH targets (e.g., *Ptch1* and *Ptch2*), as the authors do in Figure 2C.

As suggested, we have expanded the hedgehog pathway target genes considered in Figure 2b to include *Ptch1*, *Ptch2* and *Hhip*, each of which showed a similar pattern of regulation to that observed for *Gli1* and *Gli2*. When considered alongside the *Gli1* reporter luciferase studies performed, these results further strengthen our conclusion that *Slitrk5* loss-of-function enhances hedgehog pathway activation.

5. In Figure 3A, I have the same concern as in Figure 2A— why is there no difference in ALP activity between WT and *Slitrk5* mutant cells? I was expecting that the results portrayed in Figure 1 would hold true across experiments.

We agree that the lack of a basal phenotype in Figure 3a is a potential source of confusion that should be clarified. As discussed in response to point 2 above, we had deliberately chosen an early timepoint (here, day 3) in the course of osteoblast differentiation when there is minimal to no difference in ALP levels, as interpreting the results of purmorphamine stimulation could be difficult in the face of basal differences in ALP levels. To experimentally clarify this point, we have repeated this study at a later point in differentiation (day 8), finding that, as expected, there is a basal increase in ALP levels in SLITRK5-deficient cells (added as Supplementary Figure 3a). Interestingly, this difference disappears in the face of purmorphamine mediated hedgehog signaling pathway activation and the resulting enhancement of ALP acquisition. Finding that the phenotype of SLITRK5-deficient cells normalizes with saturation of hedgehog pathway activity is consistent with negative regulation of hedgehog signaling being a key mechanism by which SLITRK5 deficiency alters overall osteoblast differentiation.

6. The IP experiments in Figure 3B do not make sense to me. That is, why is there a band in the anti-FLAG input blot (lane one, top blot) in lysates that have not been transfected with SLITRK5-

FLAG? Other issues in this blot (and the other blots) include the lack of MW markers, and the lack of a loading control for the input lanes.

We thank the reviewer for identifying this issue, as this confusion was due to an error in figure labeling. We have addressed this issue while also providing the requested loading controls and molecular weight markers by conducting an independent repeat of the experiments in Figure 3b. These repeat studies confirm the results of the original figures while also improving the quality of these figures.

7. The data with Slitrk6 presented in Figure S1B, C are very interesting. However, I would have liked the authors to explore this in greater detail. For example, does Slitrk6 expression inhibit expression of HH targets? What if you expressed Slitrk1 (or another Slitrk member)? In other words, do SLITRK proteins perform overlapping functions, or is this role in HH signaling restricted to Slitrk5? The authors do not need to exhaustively explore this possibility, but it seems that with a couple straightforward overexpression experiments (and/or IP experiments), the authors could greatly improve the impact of their work (in this case, either answer is an interesting one!).

We thank the reviewer for this suggestion as we believe that addressing this has clarified an important issue in the initial submission. We performed the suggested experiment of overexpressing SLITRK6 and another SLITRK family member, SLITRK1. We found that, despite achieving efficient overexpression of both SLITRK6 and SLITRK1, neither had a similar effect to SLITRK5 to suppress hedgehog responses (Supplementary Figure 3d). We also performed co-IP experiments, finding, in contrast to results with SLITRK5, no evidence of physical interaction between SHH with either SLITRK6 or SLITRK1 (Supplementary Figure 3b, c). These results prompted us to further examine the prior SLITRK6 knockdown results. This investigation identified that both of the shSlitrk6 constructs we utilized had an unexpected ability to also reduce Slitrk5 mRNA levels despite not targeting regions of high sequence homology between Slitrk6 and Slitrk5 (See the Figure for reviewers R1). Overall, these results suggest that the hedgehog regulatory functions of SLITRK5 are likely not shared by other SLITRK family members, especially SLITRK1 and SLITRK6. To avoid confusion, we have replaced the data on the phenotypic effects of Slitrk6 knockdown with this new data on the impact of SLITRK1 and SLITRK6 overexpression on hedgehog signaling and the negative data on SLITRK1/6 interactions with SHH (Supplementary Figure 3 b-d). We have also modified our manuscript text accordingly.

8. The data presented in Figure 4 are quite interesting. However, there is no connection to changes in the level of HH signaling. If the authors could perform an assessment of HH pathway activity (e.g., Gli1 in situ hybridization or RNAscope, qPCR or western blot), that would significantly improve the connection with the previous figures.

We agree that additional in vivo evidence that the fracture healing phenotype of Slitrk5^{-/-} mice relates to alterations in hedgehog signaling identified elsewhere in the manuscript would

strengthen our findings. As suggested, we performed Gli1 in situ hybridization, finding that Gli1 levels were increased as expected in the mesenchymal component of the fracture callus (added as revised Figure 5k). Concurrent Col1a1 in situ hybridization found an increase in Col1a1 expression by osteoblasts in Slitrk5^{-/-} mice consistent with the increased callus bone formation observed. These results were also confirmed by isolating RNA from total fracture callus tissue and performing qPCR, where similar increases in Gli1 and Col1a1 were observed. As suggested by the reviewer, we believe that this data strengthens the connection between fracture healing studies and the hedgehog signaling studies elsewhere in the manuscript. During revisions, we also performed a similar analysis of expression of hedgehog target genes at baseline in the absence of fracture, finding a similar increase in Gli1, Hhip and Ptch1 in total mRNA isolated from the tibia of Slitrk5^{-/-} mice (added as revised Figure 5a).

Minor Comments:

1. In Figure 1A, D, E, as well as Figure 2A-D, the authors need to include individual data points in their histograms (as they do in Figure 4).

Thank you for this suggestion. We have converted all of the figures to display individual data points as requested.

2. In Figures 1, 2, and S1, the authors vacillate between GAPDH, HPRT and beta-actin for normalization in their qPCR experiments. It would be helpful to include a sentence of rationale for changing these controls between experiments.

Thank you for noticing this point. This change in the normalization housekeeping gene utilized reflects that this study was conducted across several years and over this time shifts in preferred normalization genes within the laboratory occurred. To improve the consistency of these figures, all of the murine qPCR studies were re-run, now consistently using GAPDH as the normalization control (added as Figure 2c). These repeat studies confirm the conclusions of the original experiments.

3. In Figure S3, the authors need to re-run their input so that the reader can assess the levels of dECD expression. This was not an issue in Figure 3C, so I am not sure why it is an issue here.

Thank you for noticing this point. This difference is likely due to Supplementary Figure 3c being performed in C3H10 cells while Figure 3c was performed in 293T cells, with a background signal found only in C3H10 lysates overlapping with the dECD band. To fix this issue, we repeated Supplementary Figure 3c in 293T cells, obtaining clean detection of input dECD expression levels. This new data is added as Figure 3h.

4. In Figures 3F vs. 3G, why is there a doublet in one FLAG blot (3G), but not the other (3F)?

We suspect that this difference reflects differences in the degree of separation occurring during the SDS page step for these two studies. Indeed, we note that careful examination of the figures in Song et al. *Developmental Cell* suggests that occasionally closely migrating doublet forms may be similarly observed. This may reflect posttranslational modification of SLITRK5; however, further investigation will be needed to make a definitive statement on this point.

5. In Figure S3, can the authors demonstrate that the deltaICD construct still interacts with SHH? If the authors can perform this experiment, and clean up their loading control, then I would recommend moving the revised figure to the primary data.

We agree that demonstrating that the SLITRK5 construct with deletion of the intracellular domain (dICD SLITRK5) retains SHH interaction is an important control. In an additional experiment added during revisions, we find that both full length and dICD SLITRK5, but not the SLITRK5 construct with deletion of the full extracellular domain (dECD SLITRK5) display robust interaction with SHH. The LRR2 domain and flanking regions in the extracellular domain of SLITRK5 are sufficient to mediate this interaction as SLITRK5 with deletion of the LRR1 domain (dLRR1) retains the ability to interact with SHH. This new experiment also overall improves the quality of SLITRK5 SHH interaction studies with cleaner, lower background data than the studies provided with the initial submission. As suggested, this data has been added to main text Figure 3c.

Reviewer #2 (Remarks to the Author):

Major points:

1. The claim that *Slitrk5* is specifically expressed in osteoblast cells is very important for the conclusion and approached they used in this study. However, there is no convincing data to show the temporospatial expression patterns of *Slitrk5* in skeletal formation. In situ hybridization is the best way to achieve this goal. Figure 1a is an RT-PCR data with no cellular resolution. For instance, it is not known whether *Slitrk* is expressed in periosteum or perichondrium or growth plates.

We agree that further demonstration of SLITRK5 expression in osteoblasts would strengthen the manuscript. Please see the response to point 2 immediately below for additional discussion.

2. The data quality of Fig. 1b is poor. It looks like that beta-galactosidase expression was everywhere in the bone marrow of both wt and *Slitrk5*^{+/-} mice. Images of the entire bone with growth plates are necessary to support the claim that *Slitrk5* is expressed specifically in the osteoblast cells. The age of the mice where the cells or femur samples are taken must be indicated too.

In response to both points 1 and 2 above, we agree that the beta-gal signal observed in osteoblasts in Figure 1b is not especially robust and that additional evidence of SLITRK5 expression in osteoblasts would strengthen the manuscript. To this end, during revisions, we have performed additional immunofluorescence studies, finding robust beta-gal staining in osteopontin⁺ osteoblasts (added as a revised Figure 1b). Additionally, low power images are provided as requested, demonstrating an absence of SLITRK5 expression in the growth plate, and overall reinforcing our claims that, within the skeleton, SLITRK5 expression is restricted to osteoblast lineage cells. Overall, we believe that this new data is clearer than the prior immunohistochemistry due to lower background of the immunofluorescence performed, thereby strengthening the manuscript.

3. Fig. 2, alteration of Hh signaling should be shown in vivo in the bone samples.

We agree that additional data more directly linking the in vivo bone phenotype observed to the hedgehog signaling functions of SLITRK5 would strengthen the manuscript. To this end, Gli1 in situ hybridization and qPCR were performed, initially focusing on analysis of the fracture callus as this is where the most robust skeletal phenotype is observed (added as Figure 5k, l). Indeed, increases in Gli1 levels in *Slitrk5*^{-/-} mice are observed consistent with the expected enhancement in hedgehog signaling. Additionally, we examined if similar findings could be observed at baseline in the absence of fracture, finding increases in Gli1, Hhip and Ptch1 mRNA levels taken from total tibial mRNA isolated from *Slitrk5*^{-/-} mice (added as revised Figure 5a). These results serve to better integrate the phenotypic analysis and the hedgehog signaling analysis performed in this study by providing evidence for the relevance of the hedgehog signaling alterations to the in vivo phenotypes observed.

4. Fig. 3F, G, it looks like Slitrk5 reduced the protein levels of PTCH1-HA levels. Is this related to Slitrk5 function in regulating Hh signaling?

We agree that in many cases that PTCH1 levels appear to be reduced when SLITRK5 is overexpressed. To investigate this further, during revisions titrated amounts of SLITRK5 were co-expressed with a GFP-PTCH1 construct (Figure for reviewers R2), finding no evidence that SLITRK5 directly regulates PTCH1 protein levels. Thus, any alteration in PTCH1 levels observed doesn't consistently appear to correlate with the amount of SLITRK5 or SLITRK5 signaling capacity.

5. It looks like that Slitrk5 did not interfere with the SHH-PTCH1 binding, as SHH regulates PTCH1 and SMOOTHED cilium localization, the cilium localization of Slitrk5 and the impact of Slitrk5 loss on cilium localization of PTCH1 and SMOOTHED should be investigated to gain mechanistic understanding of the inhibitory mechanism of Slitrk5 in Hh signaling.

We in particular thank the reviewer for this suggestion, as we believe that addressing this point has significantly strengthened the manuscript through a series of experiments added as a new Figure 4. First, we observed the distribution of SLITRK5, finding that it is present at baseline in the primary cilium (added as Figure 4a, b). SHH stimulation did not significantly increase the amount of SLITRK5 in the primary cilium. Additionally, the effect of SLITRK5 loss on the ciliary localization of PTCH1 and SMO was examined as suggested. Indeed, SMO was recruited to the cilia as expected after SHH stimulation. This recruitment was further enhanced in the absence of SLITRK5. Consistent with our model derived from purmorphamine stimulation and other studies that SLITRK5 acts downstream of PTCH1 but upstream of SMO, no alteration in PTCH1 basal localization or SHH induced PTCH1 redistribution were observed in SLITRK5 deficient cells (added as Figure 4c-f). Overall, these findings significantly strengthen the manuscript by now offering localization based evidence for the involvement of SLITRK5 in hedgehog pathway signaling that reinforces the results of the other biochemical and signaling studies performed.

6. Fig. 4A, both cortical and trabecular bone formation must be shown. Add cortical result.

We agree and have added full cortical and trabecular bone parameters in Figure 5d, e.

7. Fig. 4G, osteoblast differentiation and Hh signaling should be investigated in vivo with immunohistochemistry and qRT-PCR

We agree and have addressed this comment at the same time as the similar comment number 3 above. In short, Gli1 in situ hybridization and qPCR were performed from fracture callus

tissue, finding increases in Gli1 levels in Slitrk5^{-/-} mice consistent with the expected enhancement in hedgehog signaling. Osteoblast differentiation was enhanced as shown by increased Col1a1 expression by both in situ hybridization and qPCR. Thus, mRNA expression studies support that both hedgehog pathway activity and osteoblast differentiation are enhanced in the fracture callus of Slitrk5^{-/-} mice, consistent with our overall model.

8. Fig. S4C, both cortical and trabecular bone perimeters must be shown.

We agree and have added full cortical and trabecular bone parameters in Supplementary Figure 4d.

Reviewers' Comments:

Reviewer #1:

Remarks to the Author:

In the revised manuscript from Sun and Shin et al., the authors have experimentally addressed all of the major concerns from the initial review, and these data largely support the initial conclusions of the paper. Further, the new data extend the results presented in the original manuscript, and enhance the impact of their work. This manuscript represents an important addition to the field of Hedgehog signaling and Slitrk biology, and I fully support the publication of this work.

Reviewer #2:

Remarks to the Author:

In this revised manuscript, the authors have addressed most of my previous concerns and the manuscript has been strengthened. I only have one more point. The authors showed Gli1 expression in the fracture site, but Slitrk5 is expressed in all osteoblast and more bone formation was observed during development in the Slitrk5^{-/-} mice. Gli1 and/or other Hedgehog signaling target gene expression should be shown in situ in wild type and mutant bone.

Reviewer #2 (Remarks to the Author):

In this revised manuscript, the authors have addressed most of my previous concerns and the manuscript has been strengthened. I only have one more point. The authors showed Gli1 expression in the fracture site, but Slitrk5 is expressed in all osteoblast and more bone formation was observed during development in the Slitrk5^{-/-} mice. Gli1 and/or other Hedgehog signaling target gene expression should be shown in situ in wild type and mutant bone.

Thank you for this suggestion. As suggested, *Gli1* in situ hybridization was performed in 4-day-old wild type and *Slitrk5*^{-/-} tibial samples, finding increases in *Gli1* levels in *Slitrk5*^{-/-} mice consistent with the expected enhancement in hedgehog signaling. This new data was added as a revised Figure 5b.